# Novel Therapeutic Approaches for Cutaneous Angiosarcoma, Particularly Focusing on Immune Checkpoint Inhibitors

**DOI:** 10.3390/cancers17193163

**Published:** 2025-09-29

**Authors:** Yasuhiro Fujisawa

**Affiliations:** Department of Dermatology, Ehime University, Toon 791-0204, Japan; fujisawa.yasuhiro.fp@ehime-u.ac.jp

**Keywords:** cutaneous angiosarcoma, surgery, radiotherapy, chemotherapy, immune checkpoint inhibitors, targeted therapy

## Abstract

Angiosarcoma is a rare and highly aggressive vascular tumor with a poor prognosis, especially in patients with unresectable or metastatic disease. Systemic therapy remains the cornerstone of management in such cases. Taxane-based chemotherapy, particularly with paclitaxel or docetaxel, has been the standard first-line treatment, demonstrating moderate response rates. Molecular targeted therapies, including pazopanib and other tyrosine kinase inhibitors, have been evaluated in clinical trials but have shown limited and often transient efficacy. More recently, immune checkpoint inhibitors (ICIs) have emerged as a promising therapeutic strategy, especially in angiosarcomas arising in sun-exposed areas like the scalp and face, where tumor mutational burden tends to be higher. Early-phase trials suggest that ICIs may offer durable responses in a subset of patients. Combination strategies involving ICIs and targeted therapies are currently under investigation. Future research should focus on biomarker-driven treatment approaches and the integration of immunotherapy into standard treatment algorithms.

## 1. Introduction

Angiosarcoma is a rare, aggressive malignant neoplasm arising from endothelial cells, accounting for less than 2% of all soft tissue sarcomas [1]. Moreover, angiosarcomas show dismal prognosis with 5-year survival rates of 30% to 40% [2,3]. Cutaneous angiosarcoma (CAS), particularly of the head and neck region, is the most common clinical subtype and predominantly affects elderly individuals, especially in the elderly men [3,4]. Clinically, the tumor often presents as an ill-defined bruise-like lesion or a nodular plaque, frequently misdiagnosed as benign conditions such as hematoma, leading to significant diagnostic delays.

In addition to cutaneous forms, angiosarcoma can arise in various internal organs, including the heart, liver, spleen, and breast [5]. Primary cardiac angiosarcoma is a rare but highly lethal entity, often presenting with non-specific symptoms and rapidly progressive disease [6]. Moreover, secondary angiosarcomas may develop as a late complication of radiotherapy, particularly in breast cancer survivors [7]. While this review primarily focuses on cutaneous angiosarcoma (CAS), especially of the head and neck region, several studies referenced in the chemotherapy section included patients with angiosarcomas originating from non-cutaneous sites. Therefore, understanding the broader clinical spectrum of angiosarcoma is essential for contextualizing systemic treatment outcomes.

The pathophysiology of angiosarcoma remains poorly understood, though several risk factors have been established. These include chronic lymphedema (Stewart–Treves syndrome) [8], prior radiation therapy [9], and certain environmental exposures such as vinyl chloride and thorium dioxide. Molecular studies have revealed frequent mutations in genes such as TP53, KDR, MYC amplification (particularly in radiation-associated angiosarcoma), and alterations in angiogenesis-related pathways, including VEGF/VEGFR signaling and the MAPK cascade [10,11]. These insights have begun to inform the development of targeted therapies, although clinical translation remains limited.

Traditionally, the mainstay of treatment for localized angiosarcoma has been wide surgical excision followed with radiotherapy [12]. However, the infiltrative growth pattern, high recurrence rate, and often multicentric nature of the disease make complete resection challenging. Furthermore, a significant proportion of patients present with large tumor (>5 cm) at diagnosis, particularly in Japan [13].

Recent years have seen increased interest in systemic therapies, including conventional chemotherapy, targeted therapies, and most notably, immune checkpoint inhibitors (ICIs). Angiosarcomas are thought to be immunologically active tumors, with high tumor mutational burden (TMB) in some subsets, especially ultraviolet-induced angiosarcomas of the scalp and face [14]. Case series and early-phase trials have reported encouraging responses to ICIs, prompting further exploration in clinical trials.

In this review, we summarize novel therapeutic approaches for angiosarcoma, with a particular focus on nonsurgical strategies such as ICIs and targeted molecular therapies. While acknowledging the foundational role of surgery in early-stage disease, we emphasize the urgent need for effective systemic options to improve dismal prognosis.

## 2. Materials and Methods

A comprehensive review of the literature was conducted, including clinical trials, retrospective studies, and case series focusing on systemic treatments for advanced CAS. Therapeutic approaches covered include cytotoxic chemotherapy, molecular targeted therapies, and ICIs, as well as combination strategies. Special attention was given to biomarker studies and ongoing clinical trials. ChatGPT-4o was used for English editing purposes.

## 3. Surgery

Surgery remains the cornerstone of treatment for localized angiosarcoma. Wide local excision with negative margins is considered the standard of care in resectable cases [15]. However, unlike other soft tissue sarcomas, angiosarcomas exhibit a highly infiltrative and multifocal growth pattern, especially in the scalp and facial regions. Achieving negative margins is often difficult due to anatomic constraints and the propensity for subclinical spread. Even with aggressive surgical resection, local recurrence rates remain high, and recurrence is associated with significantly poorer survival outcomes [16].

Multiple retrospective series have emphasized the prognostic importance of margin status. For instance, Guadagnolo et al. [15] reported that patients with positive surgical margins had a significantly higher risk of local failure and distant metastases. Therefore, surgery alone is rarely sufficient and is often followed by radiotherapy to improve local control. Nevertheless, despite aggressive combined-modality approaches, outcomes remain unsatisfactory especially with large primary tumors.

Geographic differences in clinical presentation add complexity to surgical decision-making. In Japan, for example, patients often present with more advanced tumors than their Western counterparts. A nationwide Japanese survey reported that the majority of cutaneous angiosarcomas were larger than 5 cm at diagnosis, and multifocality was frequently observed [13]. In contrast, Western studies indicate a higher proportion of smaller, unifocal lesions amenable to curative resection [17]. These differences may stem from variations in access to care, patient awareness, or possibly genetic and environmental influences.

Moreover, scalp angiosarcomas—among the most common anatomical presentations—pose unique surgical challenges. The thin subcutaneous tissue overlying the scalp allows for early invasion into deeper structures such as the galea, periosteum, and even skull [4]. In such cases, wide excision may necessitate complex reconstruction using free flaps, and even then, the likelihood of residual microscopic disease remains high [18]. In elderly patients with comorbidities, extensive surgery may not be feasible, and non-surgical options must be considered.

In light of these limitations, the role of surgery in angiosarcoma should be viewed within a multidisciplinary framework. For patients with small, well-localized tumors, surgery followed by radiotherapy may offer durable control. However, in cases where surgical resection is difficult or impractical—such as those with large (>5 cm), multifocal lesions, or tumors involving anatomically complex regions like the eyelids—systemic therapies become essential.

Therefore, surgery alone is rarely sufficient and is often followed by radiotherapy to improve local control. Indeed, several studies have suggested that the addition of radiotherapy to surgery improves not only local control but also overall survival in patients with resectable cutaneous angiosarcoma [15]. In prior radiation planning reports for cutaneous angiosarcoma, treatment volumes have been defined with explicit margins. For instance, the S1 guideline for cutaneous angiosarcoma recommends a 3–5 cm safety margin around the surgical defect when irradiating skin structures [19].

## 4. Chemotherapy

Despite recent advances in molecular oncology and immunotherapy, cytotoxic chemotherapy remains a cornerstone in the treatment of unresectable or metastatic angiosarcoma. Due to the aggressive nature, high metastatic potential, and limited surgical options in advanced stages necessitate effective systemic treatment options.

### 4.1. Doxorubicin

Doxorubicin, an anthracycline antibiotic, has long been used as a standard chemotherapeutic agent in soft tissue sarcomas [20], including AS [21]. In AS, doxorubicin monotherapy has shown variable response rates, often in the range of 10–20%. A study involving soft tissue sarcomas indicated an objective response rate (ORR) of around 14% in AS subtypes treated with doxorubicin [22]. However, the cardiotoxicity associated with cumulative dosing is a major limiting factor, especially in elderly patients who represent the majority of cutaneous AS cases.

Because of its limited single-agent activity and high toxicity risk, doxorubicin is more commonly used in combination regimens or reserved for patients younger in age or ineligible for taxane therapy.

### 4.2. Paclitaxel

Paclitaxel is the most widely used chemotherapeutic agent for AS, especially in the cutaneous variant. Like docetaxel, it stabilizes microtubules and prevents mitotic progression. Although standard 3-weekly regimen is sometimes accompanied by severe bone marrow suppression and neuropathy, its weekly low-dose schedule provides a favorable safety profile, especially in elderly populations.

The first retrospective case series by Fata et al. [23] reported a response rate of 89%, demonstrating the significant activity of paclitaxel in angiosarcoma (Table 1). The ANGIOTAX study remains a landmark trial, demonstrating an ORR of 19% and a disease control rate (DCR) of 74% in patients with unresectable AS [24]. Among the Japanese population, a retrospective multicenter study in Japan showed a 1-year survival rate of over 50% in patients with cutaneous AS treated with weekly paclitaxel [25].

Paclitaxel’s antiangiogenic properties, beyond its cytotoxic effects, may be particularly advantageous in AS. This dual mechanism likely contributes to its high disease control rates. Toxicities are generally manageable, with peripheral neuropathy and myelosuppression being the most common adverse effects.

### 4.3. Docetaxel

Docetaxel promotes microtubule polymerization while inhibiting depolymerization, leading to cell cycle arrest at the G2/M phase. Although limited efficacy of docetaxel was observed in a clinical trial involving patients with sarcoma [28], docetaxel has shown some promise efficacy for angiosarcomas in smaller studies and case reports [29,30].

In clinical practice, docetaxel is sometimes considered as an alternative for patients who are intolerant to paclitaxel or when weekly administration is impractical. While paclitaxel is commonly administered on a weekly basis, docetaxel is typically given every three weeks, which may increase the risk of hematologic toxicity and limit its use in elderly or frail patients. As shown in Table 2, there were two retrospective studies focusing of using docetaxel as second-line after paclitaxel failure [31,32]. Overall response rate (ORR) of docetaxel after paclitaxel was around 20% with median PFS of 3.5–5.8 months (Table 2).

### 4.4. Eribulin

Eribulin mesylate, a drug derived from halichondrin B, shows antitumor activity by inhibiting microtubule growth without affecting shortening. This leads to irreversible mitotic blockade and subsequent apoptosis. Unlike taxanes, eribulin does not promote microtubule stabilization, which may explain its activity in taxane-resistant tumors. These mechanisms have been well summarized in previous preclinical and clinical reviews [34,35].

Case reports have also documented durable responses, highlighting eribulin’s potential as a second-line agent [36]. In a multicenter Japanese study, eribulin showed an ORR of 20.0% and a median progression-free survival (PFS) of 3.0 months in taxane-pretreated cutaneous AS patients (Table 2) [33]. Another study by Fujimura et al. [32] reported similar ORR with 20% and a median PFS of 2.8 months. Although eribulin demonstrates efficacy in taxane-resistant angiosarcoma, its clinical utility may be constrained in elderly populations due to the frequent occurrence of severe adverse events, including neutropenia and infection, as observed in prior studies.

### 4.5. Combination and Cytotoxic Alternatives

In certain cases, combination regimens incorporating doxorubicin, cisplatin, or ifosfamide have been used, although their toxicity often limits application in elderly or frail patients [37]. A case report by Lewcun et al. described two patients with breast and lower extremity AS successfully treated with a combination of doxorubicin, paclitaxel, and cisplatin [38]. While such combinations may offer a synergistic effect, their use must be carefully weighed against the potential for adverse events, particularly myelosuppression and cardiotoxicity.

Other agents, such as gemcitabine [39] or liposomal doxorubicin [40], have been trialed in small series or case reports, although their efficacy remains to be validated in larger cohorts. At present, no consensus exists regarding optimal sequencing of chemotherapeutic agents in AS, and treatment decisions are often guided by clinical factors such as age, performance status, and comorbidities.

### 4.6. Chemoradiation

Since paclitaxel or docetaxel administered concurrently with radiation not only provides radiosensitization but may also exert antiangiogenic effects, synergizing with the local effects of radiotherapy. In this context, for localized or regionally advanced cutaneous angiosarcoma (CAS), chemoradiation therapy (CRT) has emerged as a key modality, particularly in head and neck or scalp lesions where complete surgical excision is often difficult [41]. Taxane-based CRT protocols are widely utilized in Japan and have shown superior outcomes compared to surgery alone or surgery plus radiotherapy [42].

The efficacy of CRT has also been supported by prospective evidence: Fujisawa et al. conducted a multicenter retrospective study demonstrating that either paclitaxel or docetaxel combined with radiotherapy resulted in an objective response rate of 94% and a 5-year estimated overall survival rate of 56% in patients with unresectable CAS [41]. A multicenter retrospective study of 90 Japanese patients with CAS demonstrated median overall survival as ~800 days in those treated with taxane-based CRT [25]. Furthermore, Roy et al. reported outcomes of 57 patients with localized CAS treated in the United States, including 22 patients who underwent concurrent CRT with weekly paclitaxel. The CRT group achieved a significantly higher 2-year overall survival rate compared to the non-CRT group (94.1% vs. 71.6%, *p* = 0.033) [43].

These regimens are particularly advantageous in elderly patients who may not tolerate extensive surgery. Moreover, taxane-based CRT is associated with favorable local control rates and acceptable toxicity profiles, making it a compelling option for organ preservation or non-resectable disease.

Despite promising results from retrospective studies, prospective randomized trials evaluating chemoradiotherapy (CRT) for cutaneous angiosarcoma are currently lacking. Nonetheless, CRT has been widely adopted as a standard of care in many Japanese institutions and is increasingly recognized as a valuable first-line treatment option, particularly for patients with tumors in anatomically challenging locations or those who are medically unfit for surgery. Indeed, the latest Japanese clinical practice guidelines recommend CRT as a preferred first-line therapy for patients with localized disease [44].

### 4.7. Summary

Paclitaxel remains the frontline chemotherapy for advanced AS due to their established efficacy and manageable toxicity profile. Eribulin provides a promising salvage option post-taxane failure. Doxorubicin and docetaxel have more limited but still relevant roles. However, the lack of randomized controlled trials and the rarity of AS pose significant challenges in establishing standardized second-line chemotherapy protocols.

## 5. Molecular Targeted Therapy

While cytotoxic chemotherapy remains the mainstay, molecularly targeted agents have gained increasing attention due to the identification of potentially actionable pathways such as vascular endothelial growth factor (VEGF) signaling [45], *MYC* amplification [10], and aberrations in angiogenesis-related genes. Despite mixed clinical success, these therapies offer a rational and biologically informed approach to this highly vascular tumor.

The genomic landscape of angiosarcoma varies significantly by anatomical location. According to the Angiosarcoma Project led by Painter et al., cutaneous angiosarcomas, particularly those arising on sun-exposed areas such as the scalp and face, are enriched with ultraviolet (UV) signature mutations and exhibit a high tumor mutational burden (TMB), suggesting a potential sensitivity to immune-based therapies [14]. In contrast, visceral angiosarcomas—including those of the liver, spleen, and heart—often harbor genetic alterations in KDR (VEGFR2), PTPRB, and TP53, with lower TMB overall [11,14]. These molecular differences highlight the need for subtype-specific therapeutic strategies, especially when considering molecularly targeted agents or immune checkpoint inhibitors.

### 5.1. VEGF Pathway

The role of VEGF signaling in AS is well established. AS originates from endothelial cells, and overexpression of VEGF-A and its receptors (VEGFR-1, VEGFR-2) has been reported in tumor tissues, making this axis a logical therapeutic target [46]. Several agents targeting VEGF or its receptors have been evaluated.

#### 5.1.1. Bevacizumab

Bevacizumab, a monoclonal antibody against VEGF-A, was initially considered a promising agent in AS due to its antiangiogenic effects. However, clinical outcomes have been disappointing. A phase II trial by Agulnik et al. reported limited activity of bevacizumab monotherapy in AS, with an objective response rate (ORR) of 13% and median progression-free survival (PFS) of 12.4 weeks [47]. More notably, the combination of bevacizumab with paclitaxel failed to show superiority over paclitaxel alone in randomized trials [27].

#### 5.1.2. Sorafenib

Tyrosine kinase inhibitors (TKIs) that block VEGF signaling, such as sorafenib, pazopanib, and axitinib, have shown variable efficacy in AS. Sorafenib, a multikinase inhibitor of VEGFR, PDGFR, and RAF kinases, has been specifically evaluated in angiosarcoma. In the French Sarcoma Group (GSF/GETO) phase II trial, sorafenib showed limited antitumor activity in pretreated patients: in the superficial angiosarcoma (stratum A), median progression-free survival (PFS) was only 1.8 months, while in visceral angiosarcoma (stratum B), it was 3.8 months. The 9-month progression-free rates were also low—3.8% in stratum A and 0% in stratum B. Although sorafenib exhibited modest disease control (23% response in pretreated cases), the duration of benefit was short, limiting its clinical utility [48]. In a phase II trial by Maki et al., which included various sarcoma subtypes, sorafenib demonstrated limited efficacy in angiosarcoma, with few objective responses and a short median PFS of approximately 3 months [49]. Despite its mechanistic rationale, the therapeutic benefit of sorafenib appears limited, and its adverse effects, such as hand-foot syndrome, diarrhea, and hypertension, often hinder long-term administration.

#### 5.1.3. Pazopanib

Pazopanib, another VEGFR/PDGFR/c-Kit inhibitor approved for soft tissue sarcomas, demonstrated some clinical benefit in retrospective cohorts and small prospective studies in AS [50,51,52]. In a multicenter phase 2 trial (OER-073) [53], pazopanib demonstrated modest clinical activity in patients with angiosarcoma, with a disease control rate (DCR) of 51.7% and median progression-free survival (PFS) of 3.4 months. Although the objective response rate (ORR) was relatively low (6.9%), the stabilization observed in a substantial proportion of patients highlighted pazopanib’s potential disease-modifying effects, especially in those with indolent disease biology or poor performance status.

As paclitaxel remains the most widely used systemic agent for advanced angiosarcoma, it is important to clarify the efficacy of pazopanib in patients who have progressed on paclitaxel. To date, one single-arm prospective study [53] and two retrospective studies [32,54] have evaluated pazopanib in this setting (Table 3). The reported ORR ranged from 3% to 40%, with median PFS between 3.1 and 4.4 months (Table 3). These findings suggest that pazopanib may offer clinical benefit as a second- or later-line option. Although treatment-related adverse events such as hypertension and fatigue were frequently observed, they were generally manageable. To further evaluate the efficacy of pazopanib in taxane-pretreated CAS, a single-arm confirmatory trial is currently ongoing in Japan: the Japan Clinical Oncology Group (JCOG) study JCOG1605 (JCOG-PCAS protocol) [55].

### 5.2. Other Pathways

#### 5.2.1. MYC

MYC amplification is a hallmark genetic alteration in secondary angiosarcomas, particularly those induced by radiation or chronic lymphedema. In a seminal study, Manner et al. demonstrated high-level MYC amplification in such cases but not in primary angiosarcoma [56]. Additional genomic profiling by Surying et al. identified upregulation of MYC in secondary angiosarcomas compared with primary tumors, reinforcing its potential as both a diagnostic marker and a therapeutic target [57]. However, MYC remains “undruggable” due to its intrinsically disordered protein structure, and current strategies are instead focused on targeting its synthetic lethal partners or downstream effectors.

#### 5.2.2. PI3K/mTOR

Aberrant activation of the PI3K/Akt/mTOR signaling pathway has been documented in angiosarcoma. Italiano et al. examined 62 angiosarcoma cases and reported overexpression of phosphorylated S6 kinase and 4E-BP1 in approximately 42% of tumors, indicating downstream pathway activation even in the absence of PTEN deletion [58]. Chadwick et al. established a murine angiosarcoma model involving deletions of Pten, Trp53, and Ptpn12, resulting in activation of both PI3K/mTOR and MAPK pathways. Dual inhibition using mTOR and MEK inhibitors led to sustained tumor regression in this model, highlighting the potential of combined pathway blockade [59]. Although dual PI3K/mTOR inhibitors such as NVP-BEZ235 have demonstrated antiproliferative activity in preclinical studies across various solid tumors—including glioblastoma and colorectal cancer—their efficacy in angiosarcoma has not been conclusively established [60].

### 5.3. Molecular Targeted Therapy: Summary

Molecular targeted therapy in angiosarcoma offers a biologically informed yet currently limited armamentarium. Agents targeting VEGF signaling, such as pazopanib and sorafenib, have shown modest benefit, while MYC-targeted approaches remain investigational.

## 6. Immune Checkpoint Inhibitors

Recent advances in tumor immunology have introduced immune checkpoint inhibitors (ICIs) as a promising treatment for angiosarcoma, particularly CAS. Compared to angiosarcomas arising in visceral or deep soft tissue locations, CAS—particularly those located on the scalp and face—frequently demonstrates a high tumor mutational burden (TMB), suggestive of greater immunogenicity [14].

### 6.1. Tumor Mutational Burden (TMB) and Genomic Landscape

The Angiosarcoma Project, led by Painter et al., performed whole-exome sequencing and reported that CAS lesions from sun-exposed areas harbor significantly higher TMB than angiosarcomas at other anatomical sites [14]. These lesions display ultraviolet (UV)-signature mutations, often associated with increased neoantigen presentation. High TMB is considered predictive of ICI responsiveness across multiple cancers, and its presence in CAS offers a biologically plausible rationale for employing ICIs in this setting [61]. In line with this rationale, the phase II CEMangio trial investigated cemiplimab in secondary angiosarcomas, including UV-associated cases, and reported encouraging activity in this subgroup [62]. Among 18 patients, the overall response rate was 27.8%, including one complete and four partial responses. Importantly, two of the three patients with UV-signature tumors and high TMB experienced objective responses, further supporting the hypothesis that UV-induced mutagenesis may enhance ICI sensitivity in angiosarcoma.

### 6.2. Clinical Trials of ICIs

Two clinical trials provide critical insights into the efficacy of ICIs in angiosarcoma (Table 4). The first is the SWOG S1609 DART trial—a multicohort phase II basket study—evaluating nivolumab and ipilimumab in rare tumors [63]. In a cohort of angiosarcoma patients treated with immune checkpoint inhibitors (n = 16), the overall response rate (ORR) was 25%, indicating a modest level of activity. Notably, among tumors arising on the scalp or face, the ORR reached 60% (n = 5), suggesting a heightened sensitivity to immunotherapy in this anatomical subset. These findings support the hypothesis that tumor location—potentially reflecting underlying genomic characteristics such as high tumor mutational burden (TMB)—may serve as a predictive factor for ICI responsiveness.

The second is the Japanese AngioCheck trial, a multicenter phase II study assessing nivolumab monotherapy in 23 patients with histologically confirmed CAS [64]. Patients received nivolumab 480 mg every 4 weeks, with a maximum treatment period of 2 years. Although the investigator-assessed objective response rate (ORR) was 21.7% (5/23), exceeding the predefined threshold of four objective responses, central independent review confirmed only three partial responses (13.0%). As a result, the trial did not meet its primary endpoint and was considered negative.

At the time of the data cutoff, 4 patients were alive and 19 had died due to disease progression. The median overall survival was 259 days, and the median progression-free survival was 59 days. Despite the limited median PFS, graphical analyses such as the waterfall plot and swimmer plot revealed that a subset of patients who achieved stable disease experienced durable disease control over an extended period.

Tumor mutational burden was evaluated via whole-exome sequencing; among 16 evaluable cases, 7 were classified as TMB-high (≥150 mutations/MB). In the TMB-high group (n = 7), one patient achieved a partial response and three had stable disease, yielding a disease control rate (DCR) of 57.1%. In contrast, the non-TMB-high group (n = 9) included two partial responses but no cases of stable disease, resulting in a DCR of 22.2%. Although the difference did not reach statistical significance, a trend toward improved disease control was observed in the TMB-high group. These findings further highlight TMB as a clinically relevant biomarker and support its inclusion in patient selection strategies for ICI therapy.

### 6.3. Biomarkers and Predictors

While TMB has been explored as a potential predictive biomarker in CAS, other immune-related factors such as PD-L1 expression and tumor-infiltrating lymphocytes (TILs) have also been investigated. Although PD-L1 expression in CAS appears heterogeneous, its predictive value remains uncertain. In a study by Honda et al., combined infiltration of PD-1–positive lymphocytes and tumor-site PD-L1 expression was associated with favorable prognosis in cutaneous angiosarcoma [65]. Additionally, Tomassen et al. identified a UV-associated molecular subtype of angiosarcoma enriched in CD8^+^ T-cell infiltration, suggesting a T-cell–inflamed microenvironment that may favor response to immune checkpoint inhibitors [66].

Thus, combinatorial biomarker models that integrate multiple parameters—such as tumor mutational burden (TMB), tumor-infiltrating lymphocytes (TILs), and immune-related gene expression profiles including PD-1/PD-L1—may provide greater predictive accuracy for response to ICIs than any single marker alone. However, due to the rarity and heterogeneity of CAS, the development and validation of such composite biomarkers require large-scale, multicenter collaborative efforts with standardized methodologies and centralized biomarker assessment.

### 6.4. Resistance Mechanisms

Despite promising outcomes in subsets of CAS patients, a significant proportion remains unresponsive to ICIs. Proposed mechanisms of resistance include loss of antigen presentation due to beta-2 microglobulin mutations, exclusion of effector T cells from the tumor microenvironment (TME), upregulation of alternative immune checkpoints such as LAG-3 and TIM-3, and the presence of immunosuppressive myeloid cells [67]. In CAS specifically, although direct histologic evidence remains limited, emerging data from other malignancies suggest that infiltration of M2-polarized macrophages and regulatory T cells contributes to an immunosuppressive TME that can impair effective T-cell-mediated antitumor responses [68,69]. M2-like tumor-associated macrophages are known to suppress cytotoxic lymphocyte activity and promote Treg recruitment, thereby facilitating immune evasion. These mechanisms have been demonstrated in osteosarcoma and other solid tumors. Overcoming these barriers may require rational combination strategies targeting both immune checkpoints and components of the TME.

### 6.5. Future Perspective

To improve the efficacy of ICIs in CAS, ongoing studies are investigating combination therapies. Dual checkpoint inhibition (e.g., PD-1 plus CTLA-4 blockade), addition of VEGF inhibitors, and incorporation of novel agents such as CD47 or STING agonists are under early-phase evaluation [40]. Moreover, genomically matched basket trials such as NCI-MATCH may potentially include angiosarcoma patients within their rare cancer arms, facilitating a genomic-driven approach to immunotherapy. However, specific inclusion of angiosarcoma cohorts has not been formally documented in public trial data.

Precision medicine approaches leveraging comprehensive genomic and immune profiling may ultimately identify CAS subgroups most likely to benefit from ICIs. The development of adaptive trial designs for ultra-rare cancers like CAS is also essential to expedite therapeutic progress.

### 6.6. Immune Checkpoint Inhibitors: Summary

Although ICIs represent a promising therapeutic avenue for CAS, particularly in TMB-high tumors arising from sun-exposed regions, no prospective clinical trial dedicated solely to CAS has demonstrated significant efficacy of ICIs. The variability in biomarker expression and immune contexture highlights the limitations of current predictive approaches and underscores the need for a more refined, biomarker-driven strategy. Future research should prioritize integrated immune-genomic profiling, development of resistance-targeting combinations, and inclusion of CAS-specific cohorts in immunotherapy trials.

## 7. Combination Strategies

While ICIs have demonstrated clinical activity in a subset of CAS patients, durable responses remain infrequent. To enhance efficacy and overcome resistance, combination approaches—including radiotherapy (RT), TKIs, and novel immunomodulatory strategies—are increasingly under study.

### 7.1. Radiotherapy Plus ICI

RT is a foundational modality in CAS management, contributing to local disease control. Beyond cytotoxicity, RT enhances the immunogenicity of tumor cells via immunogenic cell death, release of tumor antigens, and upregulation of MHC-I molecules—thereby facilitating T-cell mediated responses when combined with ICIs [70]. In angiosarcoma specifically, a case series reported notable tumor regression in a patient treated with pembrolizumab following RT, indicating possible synergy in CAS treatment, particularly in locoregional failure or oligometastatic settings [71]. Though such evidence is anecdotal, it underscores the rationale for systematic investigation of RT-ICI scheduling, dosing, and delivery strategies in this rare disease.

### 7.2. ICI Plus TKI

CAS is an angiogenesis-dependent malignancy, typically driven by VEGF signaling. VEGF not only promotes endothelial proliferation but also contributes to immune evasion by impairing dendritic cell maturation and promoting the recruitment of myeloid-derived suppressor cells (MDSCs) and Tregs, thereby establishing an immunosuppressive tumor microenvironment. VEGF-targeting TKIs may normalize aberrant tumor vasculature, reduce infiltration of immunosuppressive cells, and enhance the efficacy of immune checkpoint inhibitors (ICIs) [72].

A notable phase II study (Alliance A091902) evaluated cabozantinib plus nivolumab in taxane-refractory angiosarcoma patients. The combination achieved an impressive ORR of 62% and median PFS of 9.6 months, signaling enhanced antitumor activity beyond monotherapy expectations [73]. These studies warrant further CAS-focused studies and suggest ICI-TKI combinations could potentiate both vascular and immune-targeted therapy.

In Japan, a phase II, single-arm, investigator-initiated clinical trial is currently underway to evaluate the combination of the TKI lenvatinib and the PD-1 inhibitor pembrolizumab in patients with advanced angiosarcoma. The results of this study (ClinicalTrials.gov Identifier: NCT06673628) are eagerly awaited and may provide further insight into the efficacy of this combinatorial strategy in the Japanese clinical setting.

### 7.3. Other Emerging Strategies

Dual checkpoint blockade (e.g., PD-1 plus CTLA-4 inhibitors) has demonstrated improved outcomes in several tumors, including angiosarcoma patients in the SWOG S1609 DART trial, who achieved a 25% response rate and sustained benefits in TMB-high cases [63]. Moreover, emerging interventions targeting innate immunity are under early investigation. STING pathway agonists, which stimulate type I interferon responses and enhance dendritic cell activity, have shown robust synergy with ICIs in preclinical models—an appealing strategy for UV-associated, TMB-high CAS, though clinical data are not yet available [74].

### 7.4. Challenges and Future Directions

Combination immunotherapy strategies in CAS face significant hurdles. The rarity of CAS impedes patient accrual, and potential additive toxicities require careful management. Additionally, predictive biomarkers to guide combination use remain underdeveloped.

To address these challenges, international collaboration via basket or umbrella trials including CAS cohorts is critical. Parallel biomarker-driven approaches—integrating TMB, PD-L1, TILs, and immune gene signatures—may enhance patient selection. Correlative studies investigating TME dynamics, angiogenic markers, and immune gene expression are also essential for understanding synergy and resistance mechanisms.

Ultimately, well-designed, biomarker-enriched clinical trials testing RT-ICI, ICI-TKI, and other novel combinations are necessary to deliver personalized, effective immunotherapeutic interventions to CAS patients.

## 8. Conclusions

CAS remains one of the most challenging malignancies in dermatologic oncology, characterized by its aggressive behavior, frequent recurrence, and poor prognosis. Over the past decade, advances in molecular biology and tumor immunology have ushered in a new era of therapeutic possibilities beyond conventional chemotherapy. Chemotherapeutic agents such as taxanes continue to serve as important option. However, their benefits are often limited in duration. The emergence of molecular targeted therapies, especially TKIs and VEGF-pathway inhibitors, has proven to be beneficial, albeit without definitive survival benefits in large-scale studies.

Among recent developments, ICIs have demonstrated the most compelling promise, particularly in CAS of the scalp and face. Clinical trials such as the SWOG S1609 DART and the Japanese AngioCheck study provide critical proof-of-concept data supporting ICI efficacy in select subpopulations. While the AngioCheck study observed partial responses in some patients, the trial did not meet its predefined primary endpoint and was therefore deemed negative. Nonetheless, these findings suggest potential benefits in biomarker-defined subsets. Biomarkers like TMB, TILs, PD-L1 and immune-related gene signatures are beginning to inform patient selection, although prospective validation remains necessary.

Combination strategies—including ICI with RT, anti-angiogenic agents, or novel immunomodulators—offer potential to overcome resistance and enhance efficacy. Integration of multi-omics profiling, precision biomarker strategies, and adaptive clinical trial designs will be vital in guiding future therapeutic development.

In summary, while the prognosis of CAS remains guarded, the expanding therapeutic landscape offers renewed hope. Continued multidisciplinary collaboration, international data sharing, and patient-centric research are essential to accelerate progress and translate scientific insights into durable clinical benefits for patients with this rare but devastating disease.

## Figures and Tables

**Table 1 cancers-17-03163-t001:** Treatment outcomes of paclitaxel.

Trials	Design	Primary Site	No. of Patients	ORR	Median PFS (Months)	Median OS (Months)
Fata 1999 [23]	Retrospective	Cutaneous	N = 9	89%	5.0	N.S.
Penel 2008 [24]	Phase 2, single arm	All sites	N = 30	18%	4.0	8.0
		Cutaneous	N = 6	N.S.		
Italiano 2012 [26]	Retrospective	All sites	N = 68	53%	4.9	8.5
		Cutaneous	N = 25	78%	8.9	20.0
Ray-Coquard 2015 [27]	Phase 2, randomized	All sites	N = 24	45.8%	6.6	19.5
Fujimura 2023 [25]	Retrospective	Cutaneous	N = 55	N.S.	N.S.	20.8

ORR: overall response ratio, PFS: progression-free survival, OS: overall survival, N.S.: not stated.

**Table 2 cancers-17-03163-t002:** Treatment outcomes of chemotherapy after paclitaxel.

Trials	Design	Treatment	No. of Patients	ORR	Median PFS (Months)	Median OS (Months)
Yonekura 2023 [31]	Retrospective	DTX	N = 6	17%	3.5	22.7
Fujimura 2023 [25]	Retrospective	DTX	N = 19	32%	5.8	12.2
		Eribulin	N = 20	20%	2.8	9.1
Fujisawa 2020 [33]	Single-arm, prospective	Eribulin	N = 25	20%	3.0	8.6

ORR: overall response ratio, PFS: progression-free survival, OS: overall survival, DTX: docetaxel.

**Table 3 cancers-17-03163-t003:** Treatment outcomes of pazopanib after paclitaxel.

Trials	Design	Treatment	No. of Patients	ORR	Median PFS (Months)	Median OS (Months)
Thiebaud 2022 [53]	Single-arm, prospective	Pazopanib	N = 29	3%	3.6	16.1
Ogata 2016 [54]	Retrospective	Pazopanib	N = 5	40%	3.1	Not stated
Fujimura 2023 [25]	Retrospective	Pazopanib	N = 11	27%	4.4	18.4

ORR: overall response ratio, PFS: progression-free survival, OS: overall survival.

**Table 4 cancers-17-03163-t004:** Clinical trials of ICIs for the treatment of cutaneous angiosarcoma.

Trials	Design	Treatment	No. of Patients	ORR	Median PFS (95%CI)	Median OS (95%CI)
Wagner 2021 [63]	Phase 2, single-arm	nivolumab + ipilimumab	All sites N = 16	25%	PFS ratio at 6 months: 38%	Not reached
			Cutaneous N = 9	60%	Not stated	Not stated
Fujisawa 2025 [64]	Phase 2, single-arm	Nivolumab	Cutaneous N = 23	13.0%	59 days (57–112)	259 days (188–387)

ORR: overall response ratio, PFS: progression-free survival, OS: overall survival, 95%CI: 95% confidence interval.

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
