# Peer review of "Novel Therapeutic Approaches for Cutaneous Angiosarcoma, Particularly Focusing on Immune Checkpoint Inhibitors"

_cancers, 2025, doi:10.3390/cancers17193163_

Round 1

Reviewer 1 Report

Comments and Suggestions for Authors

This is a well constructed and written article, and provides an excellent summary of the current status of the field of systemic therapy for cutaneous angiosarcoma.

Just a couple of minor issues: 

Line 51 correct to 'show dismal prognosis with 5-year survival rates of 30-40%'

Table 1 - include full name of DTX in legend

Tables 1 and 2 - need to include confidence intervals

Line 216 'Paclitaxel remains the frontline chemotherapy..

Line 344 >150 mutations/MB - check units

Line 387 - SARC028 did not enrol angiosarcoma, do you mean another SARC study?

Author Response

Line 51 correct to 'show dismal prognosis with 5-year survival rates of 30-40%'

>thank you for your comment and we added in the text.

Table 1 - include full name of DTX in legend

>Thank you for your comment, and we added in the legend. 

Tables 1 and 2 - need to include confidence intervals

>Thank you for your comment, and we found that most of the reported studies did not provide confidence intervals. Therefore, we deleted the confidence intervals from the table. 

Line 216 'Paclitaxel remains the frontline chemotherapy..

>Thank you for the revision, we updated the text.

Line 344 >150 mutations/MB - check units

>Thank you for your comment,  we revised the text. 

Line 387 - SARC028 did not enrol angiosarcoma, do you mean another SARC study?

Thank you for pointing this out. You are correct — angiosarcoma was not included in the SARC028 trial. We apologize for the oversight. The sentence will be revised to remove the reference to SARC028. We intended to emphasize the recent trend of basket trials incorporating rare sarcomas, including angiosarcoma in some instances, such as in exploratory arms of genomic-driven studies like NCI-MATCH. We will update the text accordingly to avoid confusion and ensure accuracy.

Reviewer 2 Report

Comments and Suggestions for Authors

The author summarizes cutaneous angiosarcoma (CAS), focusing particularly on drug therapy, especially immune checkpoint inhibitors. To improve the paper, the following additions and revisions are requested.

・This paper focuses on CAS, but "Section 4. Chemotherapy" includes treatment outcome data enrolled AS patients other than CAS. Therefore, it is desirable to briefly describe in the Introduction the existence of AS as a primary cardiac malignancy and as a secondary cancer following radiation therapy, and within that context, introduce CAS's position (proportion within the total, prognosis, etc.).
・Please add a title to Section 6.1, as it is currently missing.
・Please change the font in the main text of Section 6.2 to match the font used elsewhere in the paper.
・Section 6.2 mentions the influence of UV exposure on TMB in angiosarcoma. Considering this, please also include the Phase II immune checkpoint inhibitor trial CEMangio (https://pubmed.ncbi.nlm.nih.gov/40632032/) for secondary angiosarcoma, which included patients with a history of UV irradiation, in the discussion.

Author Response

・This paper focuses on CAS, but "Section 4. Chemotherapy" includes treatment outcome data enrolled AS patients other than CAS. Therefore, it is desirable to briefly describe in the Introduction the existence of AS as a primary cardiac malignancy and as a secondary cancer following radiation therapy, and within that context, introduce CAS's position (proportion within the total, prognosis, etc.).

>Thank you for the valuable comment. We inserted brief description in the second paragraph of the introduction section.

・Please add a title to Section 6.1, as it is currently missing.

>Thank you for the comment, we noticed that we started the section no. from 6.2. We modified the numbers in the text. 

・Please change the font in the main text of Section 6.2 to match the font used elsewhere in the paper.

>Thank you for the comment, and we revised the manuscript.

・Section 6.2 mentions the influence of UV exposure on TMB in angiosarcoma. Considering this, please also include the Phase II immune checkpoint inhibitor trial CEMangio (https://pubmed.ncbi.nlm.nih.gov/40632032/) for secondary angiosarcoma, which included patients with a history of UV irradiation, in the discussion.

>Thank you for the valuable comment. We added the description of CEMangio study in this section.

Reviewer 3 Report

Comments and Suggestions for Authors

Fujisawa reviews various treatments for cutaneous angiosarcoma.

major points

  1. For the sake of readers, standard dose and area of adjuvant irradiation should be described with appropriate references.
  2. Tables for 1st line paclitaxel and CCRT might be useful for readers as a comprehensive review.
  3. It is better to show genomic landscape of cutaneous angiosarcoma compared with visceral angiosarcoma using references such as Painter et al. and others in the target therapy section.
  4. Cutaneous AS seems to show no PI3K mutation or PTEN deletion in references such as Painter et al. The authors also should discuss about that.

Minor points

  1. P2L51 --show dismal prognosis, ranging from--  → --show dismal prognosis, with survival ranging from --
  2. P4L141 However, -- → Although standard 3-weekly regimen is sometimes accompanied by severe bone marrow suppression and neuropathy,--
  3. P5L172 Reference of reviews of eribulin (basic and clinical) should be described.

Author Response

Major points

  1. For the sake of readers, standard dose and area of adjuvant irradiation should be described with appropriate references.

       >Thank for your suggestion, we added the description in the “Surgery” section.

  1. Tables for 1st line paclitaxel and CCRT might be useful for readers as a comprehensive review.

     >Thank you for your suggestion and we added table for paclitaxel. As for CRT, we added some description in 4.6. section. 

  1. It is better to show genomic landscape of cutaneous angiosarcoma compared with visceral angiosarcoma using references such as Painter et al. and others in the target therapy section.

       >Thank you for the valuable comment, we added the description in the section.

  1. Cutaneous AS seems to show no PI3K mutation or PTEN deletion in references such as Painter et al. The authors also should discuss about that.

       >Thank you for the comment. In the original version of this article, section 5.2.2., there is a discussion of PI3K/PTEN.

Minor points

  1. P2L51 --show dismal prognosis, ranging from--  → --show dismal prognosis, with survival ranging from --

      P4L141 However, -- → Although standard 3-weekly regimen is sometimes accompanied by severe bone marrow suppression and neuropathy,--

    >Both sites were revised.

  1. P5L172 Reference of reviews of eribulin (basic and clinical) should be described.

   >We have added the description and references.